# Anaerobic Degradation of Individual Components from 5-Hydroxymethylfurfural Process-Wastewater in Continuously Operated Fixed Bed Reactors

**Muhammad Tahir Khan** [1,*], **Johannes Krümpel** [1], **Dominik Wüst** [2] and **Andreas Lemmer** [1]

[1] State Institute of Agricultural Engineering and Bioenergy, University of Hohenheim, 70599 Stuttgart, Germany; j.kruempel@uni-hohenheim.de (J.K.); andreas.lemmer@uni-hohenheim.de (A.L.)
[2] Department of Conversion Technologies of Biobased Resources, Institute of Agricultural Engineering, University of Hohenheim, 70599 Stuttgart, Germany; wuest.dominik@uni-hohenheim.de
[*] Correspondence: mt.khan@uni-hohenheim.de; Tel.: +49-711-459-24781

**Abstract:** Production of bio-based materials in biorefineries is coupled with the generation of organic-rich process-wastewater that requires further management. Anaerobic technologies can be employed as a tool for the rectification of such hazardous by-products. Therefore, 5-hydroxymethylfurfural process-wastewater and its components were investigated for their biodegradability in a continuous anaerobic process. The test components included 5-hydroxymethylfurfural, furfural, levulinic acid, and the full process-wastewater. Each component was injected individually into a continuously operating anaerobic filter at a concentration of 0.5 gCOD. On the basis of large discrepancies within the replicates for each component, we classified their degradation into the categories of "delayed", "retarded", and "inhibitory". Inhibitory represented the replicates for all the test components that hampered the process. For the retarded degradation, their mean methane yield per 0.5 gCOD was between 21.31 ± 13.04 mL and 28.98 ± 25.38 mL. Delayed digestion was considered adequate for further assessments in which the order of conversion to methane according to specific methane yield for each component from highest to lowest was as follows: levulinic acid > furfural > 5-hydroxymethylfurfural > process-wastewater. Disparities and inconsistencies in the degradation of process-wastewater and its components can compromise process stability as a whole. Hence, the provision of energy with such feedstock is questionable.

**Keywords:** biogas; biorefinery; hydrothermal carbonization; inhibition; methane; wastewater

## 1. Introduction

Fossil resources, a major source for the production of petrochemicals, are diminishing. This has prompted a gradual shift from utilizing non-renewable materials to renewable ones for the provision of bio-based chemicals and therefore reducing our dependency on scarce resources.

Out of numerous biomass-derived chemicals, 5-hydroxymethylfurfural (5-HMF) is viewed as one of the prime bio-based platform compounds. Its significance can be judged by its role as a precursor to a wide range of biochemical syntheses through chemical and biological catalysis [1]. Some of the 5-HMF derivatives, among other 170 identified downstream chemical substances, are 2,5-dimethylfuran and 5-ethoxymethylfurfural, regarded as potential biofuels candidates; adipic acid for nylon and polyester production; levulinic acid for insecticides and herbicides; and 2,5-furandicarboxylic acid as a building block for plastics, paints, and adhesives [2,3].

5-HMF can be synthesized from lignocellulosic biomass through thermochemical conversion (TCC) characterized by the pretreatment of biomass in an aqueous medium with the inclusion of a homogenous acid catalyst under high pressure and temperatures [4]. During acid hydrolysis, cellulose, hemicellulose, and lignin are converted into hexoses,

pentoses, and phenolics, respectively. 5-HMF is then generated by the dehydration of pentoses and hexoses whereas furfural solely from pentoses. Further chemical conversion of 5-HMF leads to formic and levulinic acid formation [4–8]. Commercially, 5-HMF production is achieved by employing a modified hydrothermal carbonization (HTC) process with fructose syrup as the feedstock [9]. With a bulk scale production, process-wastewater as a by-product is also generated, which is energy-rich due to the elevated hydrocarbon presence. While typical aerobic treatment systems are energy-intensive, generating an immense amount of sludge [10], treatment of process-wastewater via anaerobic digestion (AD) systems appears to be a promising alternative. AD is a microbiological process where the organic matter is converted into its most oxidized ($CO_2$) and most reduced ($CH_4$) state [11]. The produced biogas can be energetically utilized.

Investigations on the anaerobic degradability of HTC process-wastewaters from a wide range of biomasses, such as chicory roots [12], orange pomace [13], microalgae [14], maize silage digestate [15], and sewage digestate [16], were published. The findings showed that the composition of the process-wastewater was dependent upon the process parameters of the HTC and feedstock type, thus resulting in correspondingly distinct methane yields.

Wastewater generated during the TCC of biomass contains components that are known for inhibiting the anaerobic process [17]. Si et al. [18] reported 20 of such inhibiting components, originating in the hydrothermal liquefaction (HTL) aqueous phase from cornstalk. These inhibitors, when in the form of furanic and phenolic compounds, generate free radicals, disrupting enzymatic activities, weaken cell membranes, and damage genetic materials, eventually leading to cell apoptosis [19,20]. Provision of these inhibitors in low concentrations for AD can prompt gas production. However, above a definite concentration, their presence inhibits the process [21].

To the authors' knowledge, most studies for evaluating the biogas potential of such inhibitors are conducted in batch operation [22]. Hence, little to no information regarding their biodegradability in continuously operating anaerobic reactors are available. Batch mode reactors in general possess a population of mostly planktonic microorganisms. Meanwhile, in continuously operated reactors with biomass retention, they adhere to biotic or abiotic surfaces forming a biofilm that safeguards the microbial aggregates against environmental stress factors by the virtue of the extracellular polymeric substances matrix (EPS) [23–25]. This characteristic may be advantageous for the treatment of these compounds. Up-flow anaerobic sludge blanket (UASB) reactors and anaerobic filters (AFs) are common types of reactors that may be better suited for utilization for industrial and biorefinery wastewater treatment.

To ascertain the suitability of continuously operated AFs for the biorefinery's process-wastewater treatment, we utilized fully automated lab-scale AFs in the current study to investigate the biodegradability of the selected components of process-wastewater generated during 5-HMF synthesis under mesophilic temperature. These components included 5-HMF, furfural, levulinic acid, and the full (5-HMF) process-wastewater. The outcome of such a study will provide an insight into the feasibility of utilizing the process-wastewater as a feedstock on a practical scale. The properties of the components used are given in Table 1.

**Table 1.** Characteristics of the selected components of 5-hydroxymethylfurfural (5-HMF) process-wastewater utilized in this study.

| Test Component | IUPAC Name | Component Molecular Formula | Component Molecular Mass [g mol$^{-1}$] |
|---|---|---|---|
| 5-Hydroxymethylfurfural | 5-(Hydroxymethyl)furan-2-carbaldehyde | $C_6H_6O_3$ | 126.11 |
| Furfural | Furan-2-carbaldehyde | $C_5H_4O_2$ | 96.08 |
| Levunilinc acid | 4-Oxopentanoic acid | $C_5H_8O_3$ | 116.11 |
| Butyric acid (control) | Butanoic acid | $C_4H_8O_2$ | 88.11 |

## 2. Materials and Methods

### 2.1. Reactor Setup

The piping and instrumentation of the lab-scale up-flow anaerobic filters (AF) are depicted in Figure 1. Six identical AFs were utilized for the experiments. Each reactor contained bio-carriers (HX09 by Christian Stöhr, Germany) having 9 mm diameter and a surface area of 861 $m^2\ m^{-3}$, leaving a free working volume of 2.6 L and headspace of 0.7 L. The provision of heat to the AFs was realized by a water jacket, connected to a water bath system (Type E by Julabo, Germany) with an integrated pump and a thermostat. The reactors were insulated with 32 mm flexible elastomeric foam mats to ensure sustained temperature. Process parameters such as pH, redox, and temperature were measured by a combined electrode (CPS16D by Endress + Hauser, Germany). An additional sensor (TMR31 by Endress + Hauser, Germany) measured the temperature in the lower half of the AF. Substrate feeding and recirculation were carried out by 2 attached peristaltic pumps (Series 114 by Watson-Marlow, United Kingdom): P1 and P2. The feed was supplied from plastic bags connected to P1 and effluent was discharged through a siphon into a second bag of the same kind.

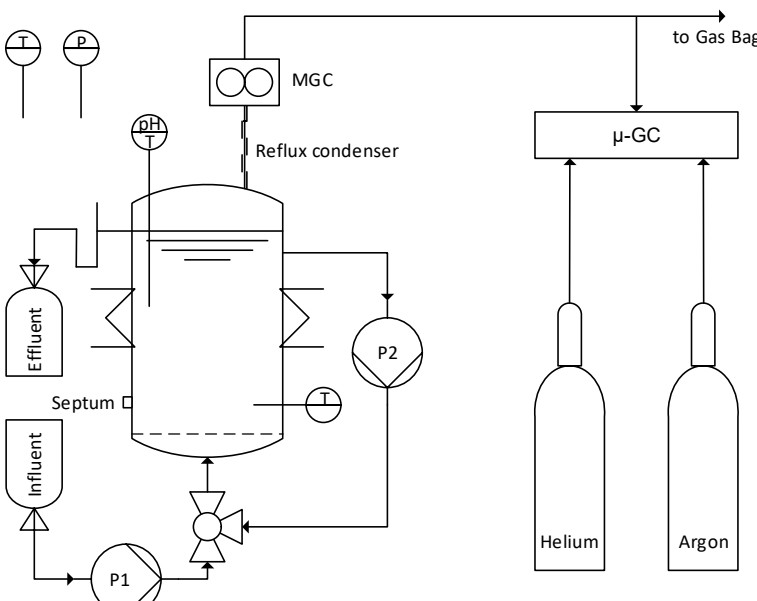

**Figure 1.** Piping and instrumentation of an anaerobic filter.

Biogas produced was passed through a mounted reflux condenser into a volumetric gas-counter (MGC-1 by Ritter Apparatebau, Germany) with a minimum resolution of 3 mL. These measurements were later recalculated to standard temperature and pressure (STP) conditions (0 °C, 1 atm) with the help of recorded pressure and temperature at the gas-counter. Following the volumetric determination, a gas chromatograph (3000 µ-GC Inficon, Switzerland) performed the qualitative analysis of the biogas every hour from each reactor through a switching valve. All process data were recorded in a relational database.

### 2.2. Analytical Methods

The chemical oxygen demand (COD) measurements of the feed substrate were performed via dilution of the samples and subsequent use of cuvette tests (Hach Lange LCK014). Prepared cuvettes were transferred to the high-performance heating block (HT 200 S by Hach Lange, Germany) for sample digestion at 170 °C for 35 min. For evaluating $Cr^{3+}$ green coloration in the sample cuvettes, owing to $K_2Cr_2O_7$ consumption by oxidizable components in the sample, we used a spectrophotometer (DR 3900 by Hach Lange, Germany) that gave the corresponding COD value.

The Inficon 3000 µ-GC performed the qualitative analysis of the produced biogas. The instrument is equipped with 2 channels, namely, channel A and channel B. Channel A analyzed the $H_2$, $O_2$, $N_2$, and $CH_4$ content of the biogas while channel B analyzed $CO_2$ and $H_2S$ content. The operating condition for the said apparatus is given in detail by Krümpel et al. [26].

Carbon along with nitrogen and the volatile fatty acid (VFA) contents of substrates were determined through Analytik Jena multi N/C 2100 S (ChD) BU and by a gas chromatograph Shimadzu GC-2010 plus, respectively. Instruments specifications, operating conditions, and methodology for carrying out such analyses are described by Lemmer and Krümpel [27].

Similarly, alcohol, sugars, lactic acid, and formic acid analysis for the feedstock were conducted by employing the Bischoff-HPLC system [26].

Techniques for the compositional analysis of the process-wastewater via Merck Hitachi Primade-HPLC system are reported by Wüst et al. [7] and Cao et al. [15].

### 2.3. Startup of the Anaerobic Filters

The reactors were operated at a temperature in the range of 42–44 °C during the entire trial course. For the startup phase, the AFs were inoculated with digestate obtained from the full-scale research biogas plant of the University of Hohenheim "Unterer Lindenhof", Eningen unter Achalm, Germany, which is fed with manure (solid and liquid) and energy crops (maize and grass silage) throughout the year.

The startup phase of the AFs was followed by feeding hydrolysate produced from grass silage in 50 L batch leach-bed acidification reactors (LBAR) operated under thermophilic conditions in the same experimental facility as described in detail by Chen et al. [28].

The AFs were operated for 5 months at a stepwise increase of the organic loading rate (OLR) up to 12 gCOD $L^{-1}$ $d^{-1}$. This maximum OLR$_{COD}$ was maintained for a week where the COD degradation reached $90.35 \pm 0.33\%$.

### 2.4. Substrate and Stock Solution Preparation

To ensure a uniform feeding regime throughout the experiment, we prepared sufficient quantities of maize hydrolysate in LBARs under the same digestion conditions as described earlier. The product was diluted with tap water. Technical grade volatile fatty acids (butyric, propionic, and acetic acid) were added, resulting in a feedstock with a COD of 21.8 g $L^{-1}$.

The prepared solution was used throughout the experiment and is subsequently referred to as "base feed". Table 2 provides an overview of the chemical composition of the base feed and the full process-wastewater. On the basis of the composition of the full process-wastewater, we chose the main components such as 5-HMF, levulinic acid, furfural, and the process-wastewater as test components for the current study. The pH values for the base feed and the process-wastewater were measured to be 4.30 and 5.10, respectively.

Stock solutions for the test components, i.e., butyric acid (Sigma Aldrich, St. Louis, MO, USA, 99% purity), 5-HMF (AVA Biochem, Muttenz, Switzerland, 99% purity), furfural (Alfa Aesar, Haverhill, MA, USA, 98% purity), and levulinic acid (Alfa Aeser, 98% purity) were prepared on the basis of Equation (1) [26], where 12.5 gCOD of an individual test component was added into a 250 mL volumetric flask and diluted with distilled water until the 250 mL volume mark. The stock solution for the full process-wastewater was prepared in a similar dilution manner. The stock solutions were stored at 5 °C afterward.

$$COD_{Component} = \frac{8(4a + b - 2c)}{12a + b + 16c} \; [gCOD \; g^{-1} C_a H_b O_c] \qquad (1)$$

Equation (1) originates from stoichiometric oxidation of the test component indicated by the succeeding reaction [29]:

$$C_a H_b O_c + \frac{1}{4}(4a + b - 2c)O_2 \rightarrow aCO_2 + \frac{b}{2}H_2O \qquad (2)$$

where *a*, *b*, and *c* are the numbers of carbon, hydrogen, and oxygen atoms in the test component, respectively.

**Table 2.** Composition of base feed and process-wastewater.

| Component/Parameter | Base Feed | | Process-Wastewater | |
|---|---|---|---|---|
| | Concentration | Carbon | Concentration | Carbon |
| | [g L$^{-1}$] | [%] | [g L$^{-1}$] | [%] |
| Formic acid | - | - | 2.66 | 2.46 |
| Acetic acid | 5.83 | 34.60 | 1.67 | 2.38 |
| Glycolic acid | - | - | 0.49 | 0.55 |
| Propionic acid | 1.56 | 11.26 | - | - |
| Lactic acid | - | - | 1.53 | 2.18 |
| Iso-butyric acid | 0.02 | 0.18 | - | - |
| Butyric acid | 5.20 | 42.11 | - | - |
| Furfural | - | - | 0.65 | 1.51 |
| Levulinic acid | - | - | 8.90 | 16.35 |
| n-Valeric acid | 0.05 | 0.46 | - | - |
| 5-Hydroxymethylfurfural | - | - | 14.79 | 30.01 |
| Caproic acid | 0.16 | 1.48 | - | - |
| Fructose | - | - | 3.87 | 5.50 |
| Unknown fractions | | 9.91 | | 39.1 |
| DOC (dissolved organic carbon) | 6.74 | 100 | 28.17 | 100 |
| TN (total nitrogen) | 0.37 | | 0.08 | |
| COD | 21.8 | | 70.15 | |

## 2.5. Experimental Procedure

During the experiments, the OLR$_{COD}$ was maintained at 3.5 gCOD L$^{-1}$ d$^{-1}$ to ensure the full capability of the AFs to degrade the additionally administered stock solutions for the experiment. This corresponded to a hydraulic retention time (HRT) of 6 days.

Out of the stock solutions, 10 mL containing 0.5 gCOD of the test component was injected into each reactor through a septum with a 3-day interval between consecutive injections. Each reactor received 4 sets of test component injections where a single set constituted 5-HMF, furfural, levulinic acid, and the process-wastewater injections. As a "check", butyric acid was injected in between each set, at the beginning and at the end of the experiment, in order to monitor the biological activity of the microbial consortia throughout the experimental phase. This schedule resulted in a total of 24 injections per substance into different reactors.

## 2.6. Analysis of Recorded Data

The data acquisition for each injection was initiated 5 h before the administration of the test components into the AFs. Through least-squares linear regression, a continuous baseline for the cumulative gas production from the base feed was determined. Subtracting the baseline from the cumulative gas production after the injection of the test component yielded the gas production curve of that test component. A detailed description for studying degradation kinetics through the given methodology is reported by Krümpel et al. [26].

Following [30,31] and on the basis of the characteristics of the gas production kinetics, we classified the conversion of the test components into methane as "delayed", "retarded", and "inhibitory", where "delayed" degradation represents methane yields between 50 to 175 mL per 0.5 gCOD, "retarded" represents 0 to 50 mL, and "inhibitory" represents no methane formation at all.

Data processing, analysis, and visualization were performed with Rstudio version 3.5.1 and Microsoft Excel 2016.

## 3. Results and Discussion

To validate the adopted methodology, we injected butyric acid into all six reactors before trial initiation. Figure 2I depicts the cumulative methane yield produced for each

hour following its administration, reaching up to $303.20 \pm 15.14$ mL CH4 gCOD$^{-1}$ after 7 h, and therefore ratifying the current methodology to investigating the biodegradability of the aforementioned test components.

Figure 2 depicts the cumulative methane yield for 1 gCOD of butyric acid, against the time (in hours) after their injection into the reactors, where I to V represent the beginning of the experiment, in between each set of test components administrations and at the end of the trial. The mean specific methane yield of the substance injection in chronological order was $303.20 \pm 15.14$ mL gCOD$^{-1}$, $283.10 \pm 13.68$ mL gCOD$^{-1}$, $280.85 \pm 21.87$ mL gCOD$^{-1}$, $307.94 \pm 10.03$ mL gCOD$^{-1}$, and $325.80 \pm 21.55$ mL gCOD$^{-1}$.

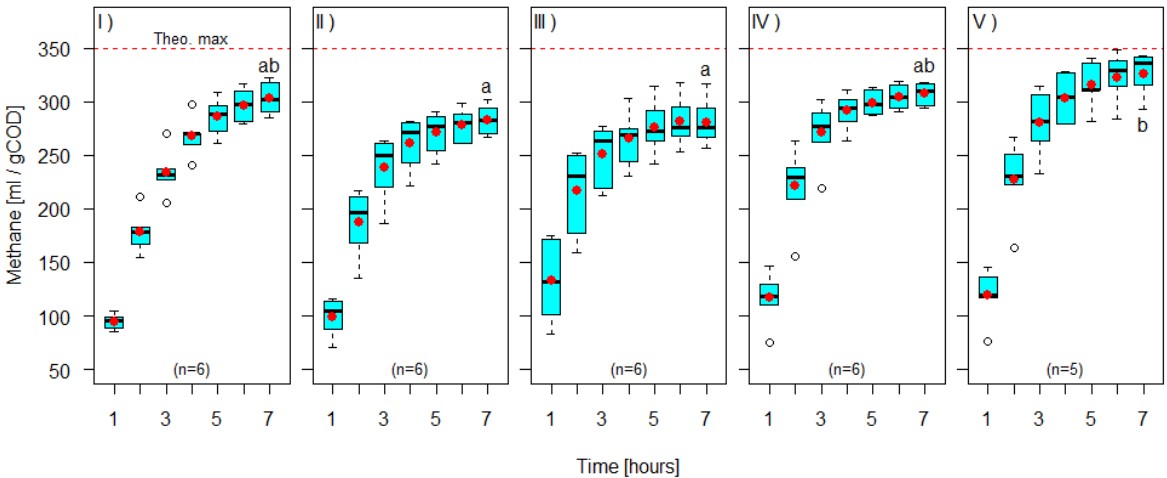

**Figure 2.** Cumulative methane production for butyric acid in a continuous anaerobic process at the given time slot. Graphs (**I–V**) represent the injection of butyric acid at the beginning of the trial, after the first set, second, third, and fourth set of test components injections, respectively. Each injection set comprised 5-HMF, furfural, levulinic acid, and the process-wastewater. The significant difference between the means for the methane yield at hour 7 is represented with different letters. The box plot is characterized by the minimum, 25% quartile, mean, median, 75% quartile, and maximum value.

A trend was observed during these runs where the methane yield dropped from $303.20 \pm 15.14$ mL gCOD$^{-1}$ to $280.85 \pm 21.87$ mL gCOD$^{-1}$, which by the end of the trial improved to $325.80 \pm 21.55$ mL gCOD$^{-1}$. Sensing stressful conditions such as antibiotics or inhibitors in their vicinity, the activities of microbial aggregates, in general, are slightly reduced initially, which they overcome by building resistance to such factors [23]. Hence, such a trend was deduced to be linked to the adaptation of microbial consortia to the test components. Moreover, with the lowest conversion of $80.24 \pm 6.25\%$ for butyric acid administered during the midway of the experiment, the microbial consortia were considered active enough to further carry our investigation on the digestion of the test components.

Upon injecting the individual components into the reactors, we observed a drop in the pH, which led to an immediate gas release for the first minutes after the injection. Krümpel et al. [26] have described this phenomenon as a result of proton provision from the introduced substance into the reactors, prompting bicarbonate to gaseous carbon dioxide transition. The amount of this immediate gas release for 1 gCOD of butyric acid administered chronologically was measured to be $14.46 \pm 2.99$ mL, $10.71 \pm 4.95$ mL, $9.98 \pm 5.87$ mL, $10.37 \pm 5.76$ mL, and $11.21 \pm 3.54$ mL.

Similar to the butyric acid, the immediate gas release for 0.5 gCOD of test components injected amounted to $7.56 \pm 3.27$ mL, $6.47 \pm 4.23$ mL, $16.75 \pm 5.12$ mL, and $6.46 \pm 2.15$ mL for 5-HMF, furfural, levulinic acid, and process-wastewater, respectively. With the subsequent base feed pumpings, the pH of the reactors was gradually regained.

As opposed to butyric acid injections, the test components did not exhibit any "normal degradation" pattern as illustrated in Figure 2, where the conversion of substance was rapid and robust. Furthermore, reproducible gas production kinetics for all the test components

could not be achieved and therefore their degradation was classified into "delayed", "retarded", and "inhibitory".

Soon after the immediate gas release process, the furanic compounds entered into an inhibitory mode producing no methane initially that lasted 2 h for 5-HMF and 1 h for furfural. The extent to which they inhibited methane baseline was evaluated to be $-5.91 \pm 8.04$ mL and $-0.30 \pm 4.33$ mL for all the observed replicates of 5-HMF and furfural, respectively. Contrary to the furanic compounds, levulinic acid and process-wastewater produced $6.10 \pm 10.45$ mL and $1.05 \pm 3.31$ mL of cumulative methane during the initial hour of their degradation, respectively.

Figure 3 represents the cumulative methane, carbon dioxide, and biogas yield for 0.5 gCOD of test components, classified as "delayed", against the time (in hours) after their injection into the reactors. On the basis of the mean measured values, we found that none of the test components reached their theoretical limits of 175 mL $CH_4$ for 0.5 gCOD. Nonetheless, the "delayed" class was considered to be adequate for further assessments in the current study. The mean methane yields per 0.5 gCOD of the administered test components by the end of 48 h were $104.37 \pm 32.52$ mL, $129.15 \pm 24.33$ mL, $137.33 \pm 33.33$ mL, and $95.62 \pm 26.20$ mL for 5-HMF, furfural, levulinic acid, and the process-wastewater, respectively. Their corresponding mean carbon dioxide release per 0.5 gCOD were measured to be $96.98 \pm 21.53$ mL, $111.08 \pm 33.68$ mL, $120.97 \pm 27.78$ mL, and $77.21 \pm 20.80$ mL, respectively. Specific gas yields, the composition of the produced biogas, and the removal percentages for the individual test components for the said class are summarized in Table 3. The order of conversion to methane according to their specific methane yields from highest to lowest is as follows: levulinic acid > furfural > 5-HMF > process-wastewater.

**Table 3.** Mean specific gas yields, mean produced biogas content, and mean percent degradation of the individual test component at 48 h after their administration into the anaerobic filters.

| Test Component | Specific Gas Yield | | Gas Quality | | Methane Conversion |
|---|---|---|---|---|---|
| | $CH_4$ | Biogas | $CH_4$ | $CO_2$ | |
| | [mL gCOD$^{-1}$] | | [%] | | [%] |
| 5-Hydroxymethylfurfural | $208.74 \pm 65.04$ | $402.30 \pm 98.40$ | $51.31 \pm 6.52$ | $48.75 \pm 6.39$ | $59.64 \pm 18.58$ |
| Furfural | $258.31 \pm 48.67$ | $479.89 \pm 84.97$ | $54.34 \pm 8.33$ | $45.76 \pm 8.73$ | $73.80 \pm 13.90$ |
| Levulinic acid | $274.67 \pm 66.66$ | $517.73 \pm 115.83$ | $52.90 \pm 3.68$ | $46.84 \pm 3.65$ | $78.47 \pm 19.04$ |
| Process-wastewater | $191.25 \pm 52.40$ | $344.76 \pm 83.21$ | $55.36 \pm 6.95$ | $45.18 \pm 6.85$ | $54.64 \pm 14.97$ |

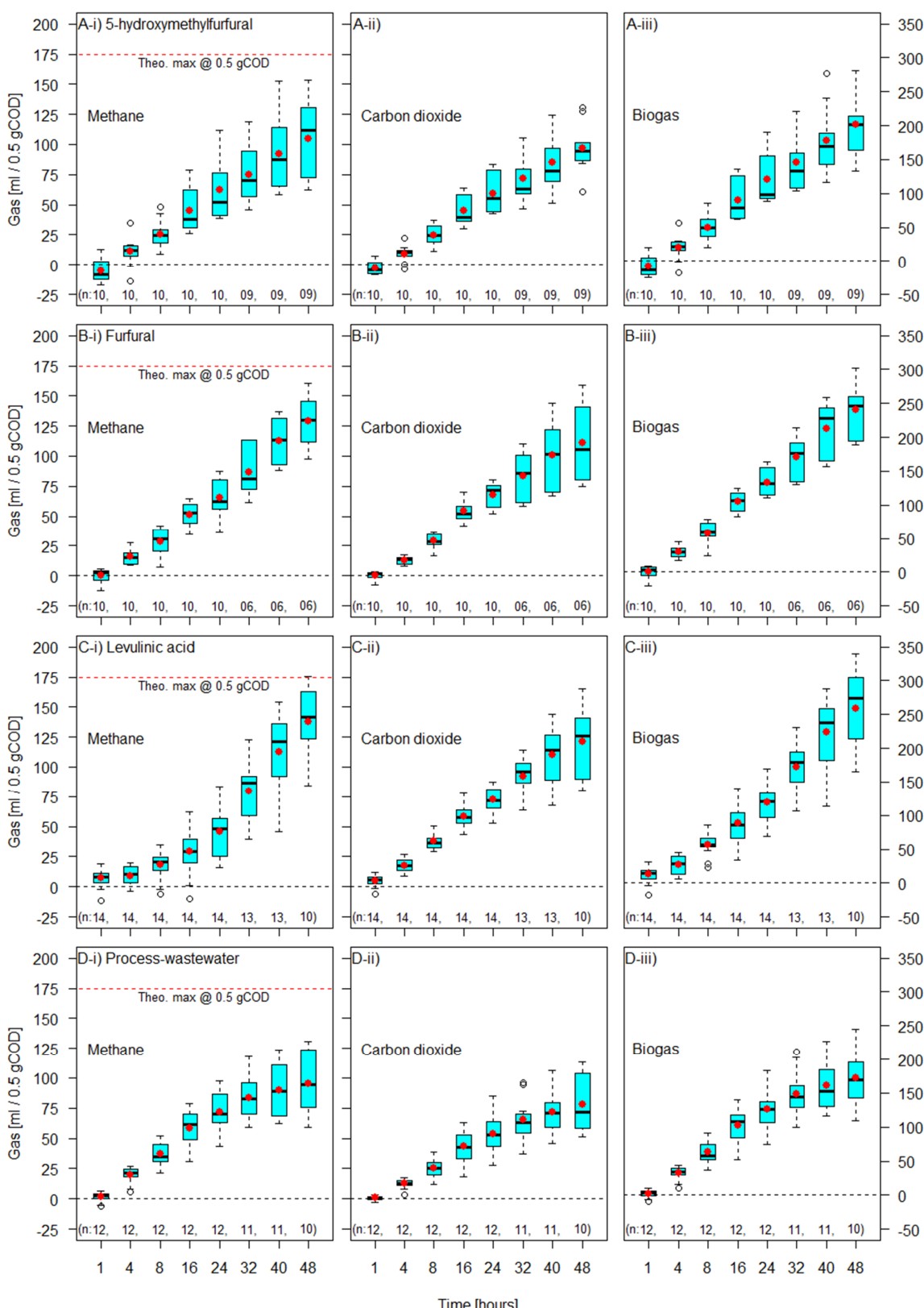

**Figure 3.** Cumulative gas production (methane (**i**), carbon dioxide (**ii**), and biogas (**iii**)) in a continuous anaerobic process at the given time slot for (**A**) 5-hydroxymethylfurfural, (**B**) furfural, (**C**) levulinic acid, and (**D**) process-wastewater. The box plot is characterized by the minimum, 25% quartile, mean, median, 75% quartile, and maximum value.

In the "delayed" classification, levulinic acid went through a longer lag phase where the lowest limit for the mean methane yield was produced at the 25th hour after its administration into the AFs. This lowest limit of 50 mL methane per 0.5 gCOD for the said classification was attained after 19, 16, and 13 h for 5-HMF, furfural, and the process-wastewater injections, respectively. As the process proceeded, large variations in the cumulative gas production and the gas production kinetics for the test components were observed, as shown in Figure 3.

Levulinic acid followed a "diauxic" criterion. VDI-4630 [30] has characterized such a pattern as a "two-phase decomposition" of the substrate. It is plausible to assume that the metabolites of the said component formed during the later process stage were readily available for the methanogens to be converted into methane than its precursors. Habe et al. [32] identified several bacterial strains responsible for the bioconversion of levulinic acid into acetic acid; propionic acid; and trehalose, a sugar with duplicate glucose molecules. Similar to the high conversion of $78.47 \pm 19.04\%$ in the current study, Park et al. [33] also observed a high removal percentage of 84.8% for 5 gCOD $L^{-1}$ of levulinic acid under mesophilic batch conditions with an increased inoculum concentration.

Likewise, the metabolites of 5-HMF and furfural are also documented to be less inhibitory than their parent material [34]. Boopathy et al. [35] and Boopathy [36] delineated the anaerobic conversion of 5-HMF and furfural into HMF alcohol (2,5-furandimethanol) and furfuryl alcohol (2-furanmethanol), respectively, via their aldehyde functional group accepting $2e^{-}$ through the coenzymes NADH (nicotinamide adenine dinucleotide, reduced form) and NADPH (nicotinamide adenine dinucleotide phosphate, reduced form) [37,38]. Hence, the presence of the hydroxyl groups in these metabolites causes them to be less inhibitory [39,40].

Furfural is described to be more inhibitory than 5-HMF owing to its lesser molar mass, and different studies have presented their outcome as such [22,33]. However, in contrast to batch mode tests, furfural was superior to 5-HMF in terms of degradability and its biogas potential in the present conditions. One cause can be related to the availability of a necessary enzyme, i.e., alcohol dehydrogenase (ADH) as HMF alcohol possesses two hydroxyl groups as compared to one for furfuryl alcohol.

The process-wastewater was the least degraded test component in this study with a removal percentage of $54.64 \pm 14.97\%$ and a mean methane yield of $191.25 \pm 52.40$ mL gCOD$^{-1}$. The low conversion can be on the account of the synergistic effect of the process-wastewater components on the anaerobic process, where the inhibition is more pronounced than the individual component alone, as the furanic compounds along with the other process-wastewater components are known to possess such effect when present in conjunction [19]. During anaerobic fermentation of the different wood types furan-containing hydrolysate for ethanol synthesis, Taherzadeh et al. [41,42] noticed a drastic reduction in the conversion of furans when their concentration exceeded 2 g $L^{-1}$, attributed to their synergistic effects on the process.

Figure 4 illustrates the cumulative methane yield for 0.5 gCOD of test components, classified as "retarded" (right) and the extent by which they inhibited the digestion process as "inhibitory" (left) against the time (in hours) after their administration into the reactors. The scales to which the test components inhibited the methane baseline after 48 h of their injection were $-69.53 \pm 73.47$ mL, $-95.06 \pm 51.78$ mL, $-35.83 \pm 55.92$ mL, and $-28.57 \pm 17$ mL for 5-HMF, furfural, levulinic acid, and the process-wastewater, respectively. Similarly, their mean cumulative methane yields for the "retarded" degradation per 0.5 gCOD after 48 h were $21.31 \pm 13.04$ mL, $28.98 \pm 25.38$ mL, $22.73 \pm 15.93$ mL, and $26.40 \pm 13.41$ mL, respectively. The number of replicates that hampered the digestion process was approximately 36% of the total for furfural when compared to the other test components ranging between 16% and 20%.

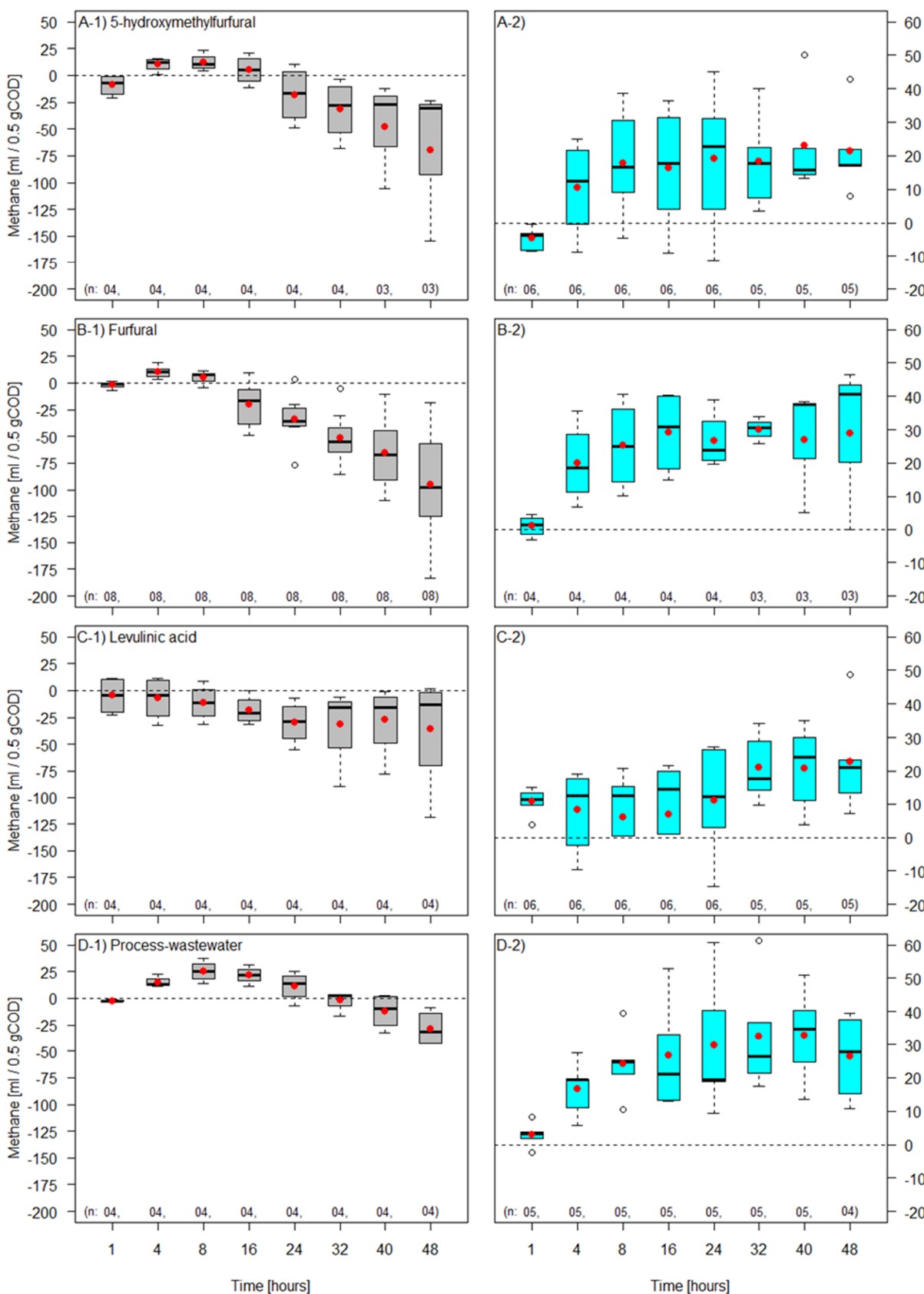

**Figure 4.** Inhibitory (**left**) and retarded degradation (**right**) for (**A**) 5-hydroxymethylfurfural, (**B**) furfural, (**C**) levulinic acid, and (**D**) process-wastewater in the anaerobic filters after its administration. The box plot is characterized by the minimum, 25% quartile, mean, median, 75% quartile, and maximum value.

The amount of the test components injected corresponding to their 0.5 gCOD were 0.33, 0.30, 0.33, and 7.13 g for 5-HMF, furfural, levulinic acid, and the process-wastewater, respectively, resulting in concentrations of 0.12, 0.11, 0.12, and 2.74 g $L^{-1}$ of the reactor's working volume. The said concentrations of furans utilized in the current study were well below their inhibition threshold documented for batch mode by Monlau et al. [22], and yet they displayed an "inhibitory" behavior and a "retarded" conversion.

As described earlier, furans are notorious for inhibiting essential fermentative enzymes and therefore hampering the degradation process. Modig et al. [43] found a reduction of more than 90% in the activity of aldehyde dehydrogenase (ALDH) and around 20% of alcohol dehydrogenase (ADH) for a given concentration of 0.25 and 0.19 g $L^{-1}$ for 5-HMF and furfural, respectively. The result of such enzyme inactivation diminishes the further metabolization of furans to their respective alcohols or acids hindering the ring-cleavage at the end [37].

Studies on the biodegradation of levulinic acid are rare, and to the authors' knowledge, complete inhibition of the anaerobic process by the said component is not reported. The retarded and inhibitory behavior of levulinic acid observed in the current study can be related to the reduced or non-availability of enzymes such as aldo-keto reductases (AKRs) to catalyze ketones [38].

## 4. Conclusions

Complexities in the anaerobic degradation of 5-HMF, furfural, levulinic acid, and the process-wastewater in a continuously operating process resulted in considerable variations in their cumulative methane yields and gas production kinetics. With added amounts as low as 0.5 gCOD, the test components every so often inhibited the process significantly. Such disparities and inconsistencies in their degradation can compromise the anaerobic process as a whole. Application of anaerobic technologies in the aspect of waste management for the biorefinery's process-wastewater preceding certain remedies might be viable, but for the provision of energy is questionable.

Additional studies are required to comprehend the biodegradability of such test components in a continuous process. In the event of persisting inhibitory behavior, perhaps proper remedial measures such as bioaugmentation, trace element supplementations, or the addition of necessary enzymes to accommodate the existing microbial consortia in the process might help achieve normal degradations of these substances.

**Author Contributions:** Conceptualization, M.T.K., J.K., and A.L.; methodology, M.T.K., J.K., and A.L.; validation: M.T.K., J.K., and A.L.; formal analysis, M.T.K., J.K., and D.W.; investigation, M.T.K., J.K., and D.W.; resources, D.W.; data curation, M.T.K.; writing—original draft preparation, M.T.K.; writing—review and editing, M.T.K. and J.K.; visualization, M.T.K., J.K., and A.L.; supervision, J.K. and A.L.; project administration, J.K.; funding acquisition, A.L. All authors have read and agreed to the published version of the manuscript.

**Funding:** The research project GRACE is funded by the Bio-Based Industries Joint Undertaking (BBI JU) under the European Union's Horizon 2020 research and innovation program under grant agreement no. 745012.

**Acknowledgments:** The authors would like to thank our project partners Stefan Krawielitzki and Gilbert Anderer from AVA Biochem BSL AG for providing the necessary resources for our current and subsequent researches.

**Conflicts of Interest:** The authors declare no conflict of interest.

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
