# Peer review of "Anaerobic Degradation of Individual Components from 5-Hydroxymethylfurfural Process-Wastewater in Continuously Operated Fixed Bed Reactors"

_processes, doi:10.3390/pr9040677_

Round 1

Reviewer 1 Report

Production of bio-based materials in biorefineries is coupled with the generation of excessive process-wastewater that requires further management. In this article 5-hydroxymethylfurfural (5-HMF) process-wastewater and its components were investigated for their biodegradability in a continuous anaerobic process. Complexities in the anaerobic degradation of 5-hydroxymethylfurfural, furfural, levulinic acid, and the process-wastewater in a continuously operating process resulted in considerable variations in their cumulative methane yields and gas production kinetics. For the retarded degradation, their mean methane yield per 0.5 gCOD was between 21.31 ± 13.04 ml and 28.98 ± 25.38 ml. The article concluded that “the provision of energy with such feedstock is questionable”. This article can be published after revision. 

In Eq. (1), notations a, b, and c should be explained, although the reader may guess its meaning.

Author Response

We, the authors of the submitted article would like to thank you for considering our work worth publishing in the said journal.

The notations "a", "b" and "c" in equation 1 are defined after revising the manuscript.

The manuscript has been subjected to English proofreading

Reviewer 2 Report

I commend all authors who contributed in this work. The manuscript addresses an interesting topic, relevant with the journal's objectives. The experimental designs, the theoretical methods and the execution of the study are adequate. The discussion of the results is clear. Generally, the study is interesting and comparable to recent studies. Therefore, it is recommended to publish in this journal.

However, to improve the article further it is advisable to review the language and to correct the following points:

  • Keywords should be in alphabetical order
  • Please improve the manuscript with a English proofreading

Author Response

We, the authors of the submitted article would like to thank you for considering our work worth publishing in the said journal.

Point 1: Keywords should be in alphabetical order

 Response 1: Keywords are placed in alphabetical order as the reviewer asked

Point 2: Please improve the manuscript with a English proofreading

Response 2: Proofreading for the manuscript’s language has been conducted.

Reviewer 3 Report

The authors proposed a paper entitled “Anaerobic Degradation of Individual Components From 5-Hy-droxymethylfurfural Process-Wastewater in Continuously Operated Fixed Bed Reactors” for the publication in Processes, MDPI.

The paper has a good scientific soundness and deserves to be published after some modifications that are required and listed in the following text.

The use of English is quite good; however, I suggest a fast revision of the manuscript, in order to correct some mistakes of syntax: some of them are reported in the following issues.

The introduction is characterized by a good description of the state of the art; however, I would cite more references, especially regarding the other process layouts used in the literature to achieve similar results.

The layout of the process used in this work has been described properly and clearly.

I suggest adding an abbreviation list, according to the guidelines of this Journal. For example: 5-HMF,  TCC, HTC, AD, HTL, EPS, UASB, AF, COD, etc.

Please, find my comments in the following. Here is the list of my other issues:

Abstract “excessive process-wastewater that requires further management.” could you quantify these volumes of water?

“ such hazardous mediums”. I think that the plural of a Latin word is “media”.

Introduction. May I suggest adding a table in which the main properties of the used acids are indicated and summarized?

“Figure 1. Piping and Instrumentation”: should you add diagram?

Even if the introduction is complete about the state of the art, the description of the aims of this paper should be put in evidence and developed.

Paragraph 2.3. “The reactors were operated at mesophilic temperatures (42-44°C)”. I would use a more simple expression such as “Reactors worked at a temperature in the range 42-44 °C…”

Pag. 5. Equation 1 is not presented nor commented properly. If I read well, CODt has not been defined as variable. “t” stands for? Please describe this equation and provide references.

Figure 2. I understand that there is a problem of space, but, in my opinion, these diagrams should have a square background area. In other words, the x and y axis should be combined in order to maintain square proportions, instead of being rectangular. The shape of these curves is obvious the same, but the proportions are different. For example, diagrams of figure 3 are clearer in terms of axis proportions, if compared with diagrams of figure 2.

Pag. 6. “A trend was observed on this occasion where…” I would refer to these runs/tests and not to “this occasion”.

I would not say “removal rate” just as percentage. Rate should somewhat like “%/time”. Or, maybe, authors could indicate it as “removal percentage”.

“resulting in per working volume” please check the syntax here.

Conclusions. I suggest expanding the point on the future perspectives of this work.

Thank you

Author Response

We would like to thank the Reviewer for providing the necessary suggestions and comments to improve our manuscript and make it sound more comprehensible to the readers. We adapted the changes in the manuscript at the places where the Reviewer asked for them. A detailed response is provided to the comments/points below, which in our opinion did not require any modifications. we hope the Reviewer will view such responses fairly.

For the "response to the comments", Please see the attachment.

Round 2

Reviewer 3 Report

Authors responded point by point to my issues, correcting typing or syntax mistakes where requested.

As an overall comment, the manuscript has improved much and now deserves to be published.

Information was correctly added to the used components, as requested.

Ok for the layout of the process as P&I diagram.

Please, cite reaction 2 in the manuscript, when presenting it.

Thank you.

Author Response

We would like to thank you for your positive comments regarding the modifications performed after revision in the manuscript and recommending it further for publication.

for the most recent comment, the response is as follow;

Point 1: Please, cite reaction 2 in the manuscript, when presenting it.

Response 1: reaction 2 is cited